# Health insurance and financial hardship in cancer survivors during the COVID-19 pandemic

Courtney P. Williams[1]*, Gabrielle B. Rocque[1], Nicole E. Caston[1], Kathleen D. Gallagher[2], Rebekah S. M. Angove[2], Eric Anderson[2], Janet S. de Moor[3], Michael T. Halpern[3], Anaeze C. Offodile, II[4], Risha Gidwani[5]

1 Department of Medicine, University of Alabama at Birmingham, Birmingham, Alabama, United States of America, 2 Patient Advocate Foundation, Hampton, Virginia, United States of America, 3 Division of Cancer Control and Population Sciences, National Cancer Institute, Rockville, Maryland, United States of America, 4 Division of Surgery, Department of Plastic Surgery, The University of Texas MD Anderson Cancer Center, Houston, Texas, United States of America, 5 Department of Health Policy and Management, UCLA Fielding School of Public Health, Los Angeles, California, United States of America

* courtneywilliams@uabmc.edu

**Data Availability Statement:** Data that support the findings of this study was collected for organizational and programmatic purposes by Patient Advocate Foundation and may be available

## Abstract

Uninsured or underinsured individuals with cancer are likely to experience financial hardship, including forgoing healthcare or non-healthcare needs such as food, housing, or utilities. This study evaluates the association between health insurance coverage and financial hardship among cancer survivors during the COVID-19 pandemic. This cross-sectional analysis used Patient Advocate Foundation (PAF) survey data from May to July 2020. Cancer survivors who previously received case management or financial aid from PAF self-reported challenges paying for healthcare and non-healthcare needs during the COVID-19 pandemic. Associations between insurance coverage and payment challenges were estimated using Poisson regression with robust standard errors, which allowed for estimation of adjusted relative risks (aRR). Of 1,437 respondents, 74% had annual household incomes <$48,000. Most respondents were enrolled in Medicare (48%), 22% in employer-sponsored insurance, 13% in Medicaid, 6% in an Affordable Care Act (ACA) plan, and 3% were uninsured. Approximately 31% of respondents reported trouble paying for healthcare during the COVID-19 pandemic. Respondents who were uninsured (aRR 2.58, 95% confidence interval [CI] 1.83–3.64), enrolled in an ACA plan (aRR 1.86, 95% CI 1.28–2.72), employer-sponsored insurance (aRR 1.70, 95% CI 1.23–2.34), or Medicare (aRR 1.49, 95% CI 1.09–2.03) had higher risk of trouble paying for healthcare compared to Medicaid enrollees. Challenges paying for non-healthcare needs were reported by 57% of respondents, with 40% reporting trouble paying for food, 31% housing, 28% transportation, and 20% internet. In adjusted models, Medicare and employer-sponsored insurance enrollees were less likely to have difficulties paying for non-healthcare needs compared to Medicaid beneficiaries. Despite 97% of our cancer survivor sample being insured, 31% and 57% reported trouble paying for healthcare and non-healthcare needs during the COVID-19 pandemic, respectively. Greater attention to both medical and non-medical financial burden is needed given the economic pressures of the COVID-19 pandemic.

upon request. Participants of this study did not agree for their data to be shared publicly. Use of data from Patient Advocate Foundation for analytic purposes requires a data use agreement that is reviewed by the external compliance officer prior to signature by both parties. Please contact PAF Compliance Officer Stephanie Trunk (Stephanie. trunk@patientadvocate.org) with inquiries.

**Funding:** The author(s) received no specific funding for this work.

**Competing interests:** I have read the journal's policy and the authors of this manuscript have the following competing interests: Dr. Rocque is supported by an American Cancer Society Mentored Research Scholar Grant (MRSG-17-051-01-PCSM) and has received research funding from Genentech, Pfizer, and Carevive and consulting fees for Genentech and Pfizer for work unrelated to the current study. This does not alter our adherence to PLOS ONE policies on sharing data and materials.

## Introduction

Despite an increase in insurance coverage rates due to the passage of the Patient Protection and Affordable Care Act (ACA) [1], rates of underinsurance remain high among Americans with cancer [2]. Underinsurance, often characterized by spending >10% of household income on healthcare, is associated with cancer treatment delays and financial distress [3, 4]. Job lay-offs and economic insecurity stemming from the COVID-19 pandemic are likely to increase rates of uninsurance and underinsurance [5]. For cancer survivors, these increases may be associated with financial hardship and result in forgone healthcare or non-healthcare purchases, such as food, housing, or utilities. This study examines the relationship between health insurance coverage and challenges paying for healthcare and non-healthcare needs during the COVID-19 pandemic in adults with cancer.

## Materials and methods

This analysis used secondary, cross-sectional survey data from individuals who had previously received case management or financial aid from Patient Advocate Foundation (PAF), a non-profit organization that helps individuals with chronic illnesses access recommended care. Internet surveys were fielded from May 20 to July 12, 2020. Survey participation was incentivized via drawings for six individual $25 gift cards. Respondents provided written informed consent for all PAF survey communications. The University of Alabama at Birmingham Institutional Review Board approved this secondary analysis of the PAF survey data. Respondents reported challenges paying for healthcare and non-healthcare needs using the single-item survey question, "Have you had trouble paying for any of the following since the beginning of the COVID-19 pandemic?" Respondents were then asked to select all that applied, which included food, household supplies, housing (rent or mortgage), utilities, phone, internet/data, car/gas/transportation, childcare/eldercare/home health services, and healthcare/medical costs (e.g., prescription medications, doctor's visits, clinical/hospital services, medical supplies).

Associations between insurance coverage and challenges affording healthcare or non-healthcare needs during the COVID-19 pandemic were estimated using adjusted relative risks (aRR) and 95% confidence intervals (CI) from modified Poisson models with robust standard errors [6]. Use of modified Poisson regression with sandwich errors for analysis of binary outcomes produces reliable estimates of relative risk, which is more easily interpretable than odds ratios produced by logistic regression [6]. Models were adjusted for age, sex, race and ethnicity, household income, education, employment, rurality, cancer type, and comorbidity count.

## Results

Surveys were e-mailed to 15,857 PAF clients and completed by 4,108 (26% response rate; S1 Table). Of 1,437 respondents reporting a previous cancer diagnosis, 38% were aged <55 years, 23% were Black or African American, 74% had annual household incomes <$48,000, and 40% had a bachelor's degree or higher (Table 1). Breast cancer was most common among respondents (35%), and 41% reported ≥3 additional comorbidities. Most respondents were enrolled in Medicare (48%), 22% in employer-sponsored insurance, 13% in Medicaid, 6% in an ACA plan, and 3% were uninsured.

Overall, 31% of respondents reported challenges paying for healthcare during the COVID-19 pandemic (Fig 1). In adjusted models, respondents who were uninsured (aRR 2.58, 95% CI 1.83–3.64), enrolled in an ACA plan (aRR 1.86, 95% CI 1.28–2.72), employer-sponsored insurance (aRR 1.70, 95% CI 1.23–2.34), or Medicare (aRR 1.49, 95% CI 1.09–2.03) were more likely to report trouble paying for healthcare compared to Medicaid enrollees (Table 2).

**Table 1. Respondent sociodemographic and clinical characteristics (N = 1437).**

| | Total N = 1437 |
|---|---|
| **Age** | |
| 19–35 | 60 (4.2) |
| 36–55 | 490 (34.1) |
| 56–75 | 790 (55.0) |
| $\geq$ 76 | 97 (6.8) |
| Sex | |
| Male | 404 (28.1) |
| Female | 1033 (71.9) |
| Race and ethnicity | |
| White | 851 (59.2) |
| Black/African American | 336 (23.4) |
| Hispanic/Latino | 120 (8.4) |
| Other | 76 (5.3) |
| Unknown | 54 (3.8) |
| Education | |
| Less than high school | 41 (2.9) |
| High school degree | 321 (22.3) |
| Some college | 501 (34.9) |
| $\geq$ Bachelor's degree | 569 (39.6) |
| Unknown | 5 (0.4) |
| Employment status | |
| Employed | 307 (21.4) |
| Retired | 397 (27.6) |
| Disabled | 535 (37.3) |
| Unemployed/other | 198 (13.8) |
| Household income | |
| $\leq$$47,999 | 1061 (73.8) |
| >$47,999 | 359 (25.0) |
| Unknown | 17 (1.2) |
| Rural-Urban Commuting Area | |
| Urban | 1083 (75.4) |
| Rural | 137 (9.5) |
| Unknown | 217 (15.1) |
| Cancer type | |
| Breast | 497 (34.6) |
| Genitourinary | 83 (5.8) |
| Gynecological | 38 (2.6) |
| Gastrointestinal | 75 (5.2) |
| Hematologic | 429 (29.9) |
| Other | 315 (21.9) |
| Comorbidity count* | |
| 0 | 583 (40.6) |
| 1–2 | 482 (33.5) |
| $\geq$3 | 372 (25.9) |
| Health insurance status | |
| Employer-sponsored | 322 (22.4) |
| ACA | 90 (6.3) |

(*Continued*)

**Table 1.** (Continued)

|  | Total N = 1437 |
|---|---|
| **Age** | |
| Medicare | 694 (48.3) |
| Medicaid | 185 (12.9) |
| Uninsured | 49 (3.4) |
| Other / unknown | 97 (6.8) |

*Count of comorbidities other than cancer

ACA = Affordable Care Act

Challenges paying for non-healthcare needs during the COVID-19 pandemic were reported by 57% of respondents, with 40% reporting trouble paying for food, 32% for housing, 28% for transportation, and 20% for internet (Fig 1). In adjusted models, respondents enrolled in Medicare (aRR 0.76, 95% CI 0.68–0.84) or employer-sponsored insurance (aRR 0.78, 95% CI 0.69–0.87) were less likely to report challenges paying for non-healthcare needs compared to Medicaid enrollees (Table 2). In adjusted models of specific non-healthcare needs, Medicare beneficiaries and respondents enrolled in employer-sponsored plans were less likely to report trouble paying for food, household supplies, housing, utilities, a phone, internet or data, and transportation compared to Medicaid beneficiaries (S2 Table).

## Discussion

Individuals with cancer are dealing with many financial challenges potentially associated with the COVID-19 pandemic, with 57% of our sample reporting trouble paying for non-healthcare needs and 31% reporting trouble paying for healthcare. Our results suggest the financial

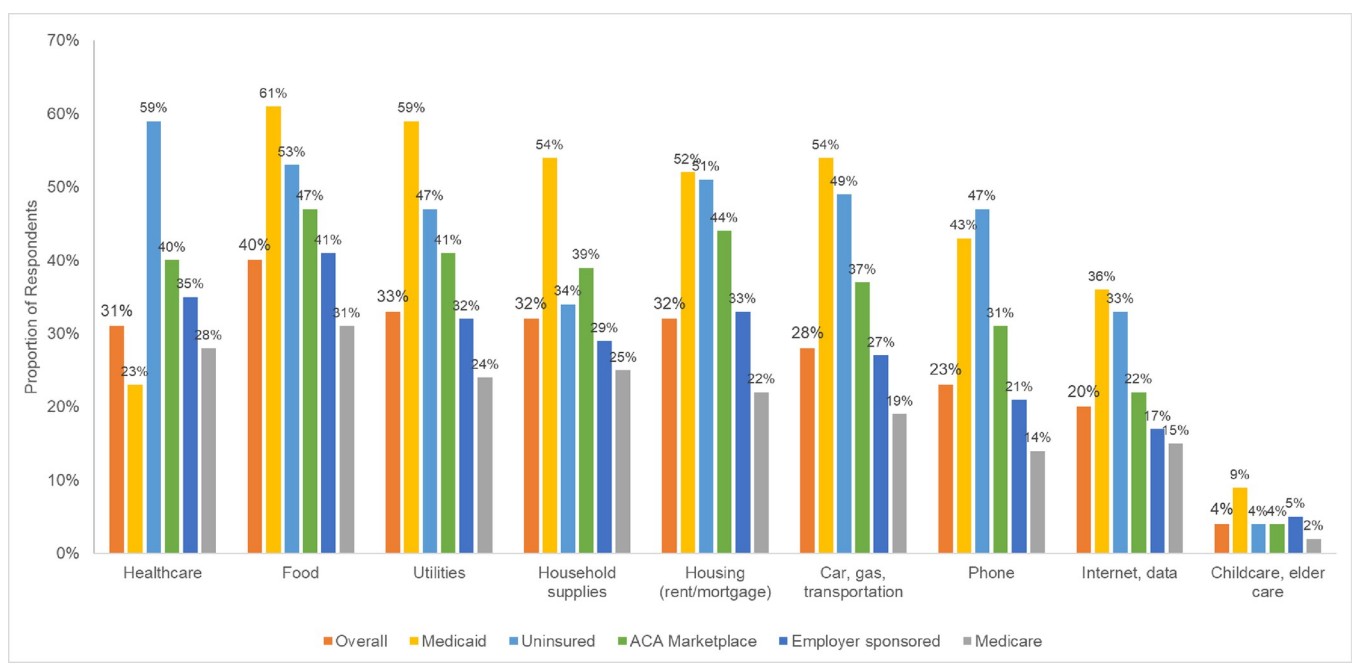

**Fig 1. Proportion of respondents reporting trouble paying for healthcare and non-healthcare needs during the COVID-19 pandemic by health insurance status (N = 1437).**

**Table 2. Adjusted model results estimating relative risk of trouble paying for healthcare or any non-healthcare need in cancer survivors (N = 1437).**

| | Trouble paying for healthcare | Trouble paying for any non-healthcare need |
| --- | --- | --- |
| | Relative Risk | Relative Risk |
| | (95% Confidence Interval) | (95% Confidence Interval) |
| Health insurance status | | |
| Medicaid | Ref. | Ref. |
| Affordable Care Act | 1.86 (1.28–2.72) | 0.88 (0.75–1.03) |
| Employer-sponsored | 1.70 (1.23–2.34) | 0.78 (0.69–0.87) |
| Medicare | 1.49 (1.09–2.03) | 0.76 (0.68–0.84) |
| Uninsured | 2.58 (1.83–3.64) | 0.91 (0.77–1.07) |
| Other / unknown | 1.44 (0.96–2.18) | 0.83 (0.71–0.98) |
| Age | | |
| 19–35 | Ref. | Ref. |
| 36–55 | 1.23 (0.82–1.85) | 0.98 (0.83–1.16) |
| 56–75 | 1.14 (0.75–1.73) | 0.83 (0.70–0.99) |
| $\geq 76$ | 1.02 (0.57–1.82) | 0.61 (0.42–0.89) |
| Sex | | |
| Male | Ref. | Ref. |
| Female | 0.96 (0.78–1.19) | 1.04 (0.92–1.17) |
| Race and ethnicity | | |
| White | Ref. | Ref. |
| Black/African American | 0.99 (0.82–1.19) | 1.31 (1.20–1.43) |
| Hispanic/Latino | 1.19 (0.91–1.56) | 1.16 (1.02–1.32) |
| Other | 1.14 (0.83–1.57) | 1.00 (0.82–1.22) |
| Education | | |
| Less than high school | Ref. | Ref. |
| High school degree | 1.28 (0.71–2.30) | 0.91 (0.74–1.11) |
| Some college | 1.44 (0.81–2.55) | 0.93 (0.76–1.14) |
| $\geq$ Bachelor's degree | 1.37 (0.77–2.45) | 0.86 (0.70–1.06) |
| Employment status | | |
| Employed | Ref. | Ref. |
| Retired | 0.81 (0.61–1.09) | 0.58 (0.47–0.70) |
| Disabled | 1.13 (0.90–1.41) | 1.13 (1.00–1.27) |
| Unemployed/other | 1.21 (0.94–1.56) | 1.08 (0.95–1.23) |
| Household income | | |
| $\leq$ \$47,999 | Ref. | Ref. |
| > \$47,999 | 1.00 (0.83–1.21) | 0.89 (0.80–1.00) |
| Rural-Urban Commuting Area | | |
| Urban | Ref. | Ref. |
| Rural | 1.25 (0.99–1.58) | 0.93 (0.80–1.07) |
| Cancer type | | |
| Breast | Ref. | Ref. |
| Genitourinary | 0.71 (0.43–1.16) | 0.84 (0.63–1.11) |
| Gynecological | 1.10 (0.74–1.66) | 0.89 (0.67–1.17) |
| Gastrointestinal | 0.87 (0.59–1.29) | 1.00 (0.84–1.18) |
| Hematologic | 0.78 (0.63–0.98) | 1.02 (0.91–1.14) |
| Other | 1.06 (0.86–1.30) | 0.99 (0.89–1.11) |
| Comorbidity count* | | |
| 0 | Ref. | Ref. |

*(Continued)*

**Table 2.** (Continued)

| | Trouble paying for healthcare | Trouble paying for any non-healthcare need |
| --- | --- | --- |
| | Relative Risk | Relative Risk |
| | (95% Confidence Interval) | (95% Confidence Interval) |
| 1–2 | 1.10 (0.90–1.33) | 1.05 (0.94–1.16) |
| ≥ 3 | 1.50 (1.24–1.82) | 1.30 (1.18–1.44) |

*Count of comorbidities other than cancer

hardship experienced by cancer survivors may have been exacerbated during the COVID-19 pandemic when compared to pre-pandemic estimates, such as those in a recent study by Han and colleagues. Using the 2016 Medical Expenditures Panel Survey data, Han estimated 16% of cancer survivors experienced material financial hardship, which includes reduced spending on non-healthcare needs, and 27% experienced delayed or forgone health care due to cost [7]. Our study also showed risk of payment challenges differed by insurance coverage status. Medicaid beneficiaries were at lowest risk of challenges paying for healthcare, but highest risk of trouble affording non-healthcare needs. Conversely, respondents enrolled in an ACA plan, employer-sponsored insurance, or Medicare were more likely to report difficulties in paying for healthcare compared to Medicaid enrollees.

Our results suggest private insurance and Medicare coverage may not sufficiently protect against out-of-pocket medical costs in individuals with cancer. In our study, individuals with ACA, employer-sponsored, or Medicare insurance were at higher risk of trouble paying for healthcare during the COVID-19 pandemic compared to Medicaid enrollees. This is likely due to substantial cost sharing requirements posed by these insurance mechanisms. An estimated 47% of privately-insured patients are enrolled in high-deductible health plans [8]. In 2020, individual silver plan ACA deductibles averaged $4,450 and 26% of individuals with employer-sponsored insurance had annual deductibles of at least $2,000 [9, 10]. After deductibles are met, normal patient cost-sharing provisions such as copayments or coinsurance apply. Though the majority of Medicare beneficiaries are enrolled in supplemental coverage which offsets beneficiary Medicare Parts A and B cost sharing, Medicare Part D, which covers many oral anticancer medications, has no cap on patient out-of-pocket costs and imposes 25% co-insurance during the initial coverage period and in the coverage gap. Because the mean anticancer medication price was almost $14,000 in 2018, this benefit design commonly results in high out-of-pocket spending for enrollees with a cancer diagnosis [11]. Conversely, out-of-pocket costs for premium and cost sharing are capped at 5% of household income for all Medicaid enrollees [12]. Efforts towards increasing provider awareness of potential financial hardship in insured patients is needed during treatment and survivorship care planning.

On the other hand, over half of our study respondents reported trouble paying for non-healthcare needs during the COVID-19 pandemic, with similar risks of trouble found in respondents who were uninsured, ACA-insured, and enrolled in Medicaid. Difficulties in paying for non-medical necessities can affect cancer outcomes. Food-insecure cancer survivors have higher odds of forgone, delayed, or altered cancer treatment than those who are food secure [13]. Challenges paying for transportation, cited by almost one-third of our study respondents and half of Medicaid enrollees, can compound quickly and negatively impact receipt of care [14]. The pandemic-induced shifts to telehealth may not sufficiently offset access challenges stemming from transportation, as 53% of respondents in our study reporting transportation cost challenges also reported trouble affording internet [15]. While addressing food or housing insecurity may extend beyond the immediate scope of clinical practice,

provider inquiry and accommodation of patient transportation and internet challenges, such as scheduling same-day appointments with cancer care team members, could reduce financial burden and improve receipt of recommended care. Of note, ACA-insured and Medicaid patients reported similar levels of difficulty in paying for non-healthcare necessities. However, there is no differentiation between ACA or employer-sponsored insurance in the electronic medical record. Patients with either type of coverage will appear as privately insured. Thus, providers should be aware that many of their privately insured patients face the same financial challenges as their Medicaid patients, and may wish to extend any discussions of financial burden or assistance to their privately insured patients as well.

The results of our study should be considered within limitations. The survey captures data from cancer survivors who previously received help accessing or paying for care from PAF and may not be representative of the larger cancer patient population. Information potentially associated with health insurance coverage and challenges paying for healthcare and non-healthcare needs, such as time since cancer diagnosis, current health care use or needs, and more detailed demographic information, was limited by use of secondary data. The results of our study could have been impacted by pandemic-related employment changes. However, this was uncommon in our study with only 2% of respondents reporting employment loss due to the COVID-19 pandemic. We had a low survey response rate, potentially due to the COVID-19 pandemic, which may result in selection bias. Our sample may also be biased towards individuals able to navigate services from a non-profit organization or access web-based surveys.

## Conclusions

In this sample of cancer survivors, 31% and 57% reported trouble paying for healthcare and non-healthcare needs during the COVID-19 pandemic, respectfully. Those with private insurance or Medicare were more likely to report trouble paying for healthcare, while those enrolled in Medicaid most often reported trouble paying for non-healthcare needs. System-level efforts towards ensuring insurance coverage adequately addresses healthcare needs and provider-level efforts to incorporate financial information in clinical decision-making could aid in reducing patient-level financial hardship associated with the COVID-19 pandemic.

## Supporting information

**S1 Table. Non-respondent sociodemographic characteristics (N = 11,749).**
(DOCX)

**S2 Table. Adjusted model results estimating relative risk of trouble paying for non-healthcare necessities in under-resourced cancer survivors (N = 1,437).**
(DOCX)

## Acknowledgments

**Previous presentations:** This work was presented as a poster presentation at the virtual 2021 Academy Health Annual Research Meeting, June 14–17, 2021.

## Author Contributions

**Conceptualization:** Courtney P. Williams, Gabrielle B. Rocque, Kathleen D. Gallagher, Rebekah S. M. Angove, Risha Gidwani.

**Data curation:** Courtney P. Williams, Gabrielle B. Rocque, Nicole E. Caston, Kathleen D. Gallagher, Rebekah S. M. Angove, Eric Anderson.

**Formal analysis:** Courtney P. Williams, Nicole E. Caston, Risha Gidwani.

**Investigation:** Courtney P. Williams, Gabrielle B. Rocque, Nicole E. Caston, Eric Anderson, Janet S. de Moor, Michael T. Halpern, Anaeze C. Offodile, II.

**Methodology:** Courtney P. Williams, Nicole E. Caston.

**Project administration:** Rebekah S. M. Angove.

**Resources:** Gabrielle B. Rocque, Rebekah S. M. Angove, Eric Anderson.

**Supervision:** Gabrielle B. Rocque, Kathleen D. Gallagher.

**Validation:** Courtney P. Williams, Risha Gidwani.

**Writing – original draft:** Courtney P. Williams.

**Writing – review & editing:** Courtney P. Williams, Gabrielle B. Rocque, Nicole E. Caston, Kathleen D. Gallagher, Rebekah S. M. Angove, Eric Anderson, Janet S. de Moor, Michael T. Halpern, Anaeze C. Offodile, II, Risha Gidwani.

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
