## [Decision Letter · Decision Letter 0]

25 Jan 2022

PONE-D-21-32975Health insurance and financial hardship in cancer survivors during the COVID-19 pandemicPLOS ONE

Dear Dr. Williams,

Thank you for submitting your manuscript to PLOS ONE. After careful consideration, we feel that it has merit but does not fully meet PLOS ONE’s publication criteria as it currently stands. Therefore, we invite you to submit a revised version of the manuscript that addresses the points raised during the review process.

We look forward to receiving your revised manuscript.

Kind regards,

Raymond Nienchen Kuo, Ph.D

Academic Editor

PLOS ONE

Journal Requirements:

a) Did participants provide their written or verbal informed consent to participate in this study?

3. "Thank you for stating the following in the Competing Interestsdelete as section: 

"I have read the journal's policy and the authors of this manuscript have the following competing interests: Dr. Rocque is supported by an American Cancer Society Mentored Research Scholar Grant (MRSG-17-051-01-PCSM) and has received research funding from Genentech, Pfizer, and Carevive and consulting fees for Genentech and Pfizer."

We note that you received funding from a commercial source: Genentech

Reviewers' comments:

Reviewer's Responses to Questions

**Comments to the Author**

1. Is the manuscript technically sound, and do the data support the conclusions?

Reviewer #1: Yes

Reviewer #2: Yes

2. Has the statistical analysis been performed appropriately and rigorously? 

Reviewer #1: Yes

Reviewer #2: Yes

3. Have the authors made all data underlying the findings in their manuscript fully available?

Reviewer #1: Yes

Reviewer #2: Yes

4. Is the manuscript presented in an intelligible fashion and written in standard English?

Reviewer #1: Yes

Reviewer #2: Yes

5. Review Comments to the Author

Reviewer #1: This is a well written and important topic on financial burden for cancer patients in the United States during the pandemic. Overall this is a well thought out paper, but I do have a few significant suggestions, and one minor comment.

Significant

1. The authors have made no attempt to identify how the financial impact to patients has changed since the pandemic. I am not suggesting additional analysis, but rather an examination of data pre-pandemic to see if the impact is different (or the same) as it was pre-pandemic in the discussion or limitations.

2. The authors have not addressed the overall impact on employment during the pandemic. Although some federal funding was provided in many cases this would have been a decrease over their regular income stream. Again I am not suggesting additional analysis but at a minimum this needs to be highlighted in the limitations, or an examination of the literature on this topic should be included in the discussion.

Minor

pg 12 of 21 line 122 "which covers may oral..." should read "which covers many oral..."

Reviewer #2: This is a well-written and concise analysis, and it provides a meaningful addition to the growing (and necessary) body of literature on the impact of COVID-19 on cancer patients and survivors. I particularly appreciate your including cost of internet in your survey, as this is, as you note, a potential barrier to telehealth uptake and may limit survivors' ability to engage with other aspects of the healthcare system as well. My comments are minimal but include the following:

1) Please consider omitting your use of "under-resourced" throughout the manuscript, or at least provide a clear definition to understand the research team's characterization of this term.

2) Did you collect any information on time since treatment or diagnosis, or on the respondents' current healthcare use/needs? If so, please report; if not, please note as a limitation, as these factors, particularly current healthcare needs, may influence one's difficulty paying for healthcare use.

3) Please provide more detail on how your outcomes of interest were measured? Were these yes/no questions, or did you use a scale? If the latter, provide detail on how the scale was dichotomized.

4) Was age collected as a continuous variable, or was it collected in the categories reported in Table 1? If the former, consider revising your categorizations more meaningfully to reflect age 65 as the age of Medicare eligibility and 18-39 as the NCI's definition of young adult. Similarly, how was the income threshold of $48K determined?

6. PLOS authors have the option to publish the peer review history of their article (what does this mean?). If published, this will include your full peer review and any attached files.

Reviewer #1: No

Reviewer #2: No

---

## [Author Response · Author response to Decision Letter 0]

25 Apr 2022

February 21, 2022

Raymond Nienchen Kuo, PhD, Academic Editor

PLOS ONE

Manuscript title: Health insurance and financial hardship in cancer survivors during the COVID-19 pandemic

Dear Dr. Kuo and Reviewers,

We wish to thank the reviewers for their thoughtful comments and for the opportunity to respond. Please see our response below and the updated manuscript for our responses to the reviewer’s suggestions.

Thank you again for your consideration of this manuscript.

Response to journal requirements:

Response: We have edited our manuscript to meet the PLOS ONE style requirements.

a) Did participants provide their written or verbal informed consent to participate in this study?

Response: Survey respondents provided electronic, written consent for their participation in this survey. We have clarified this in the methods section.

“Respondents provided written informed consent for all PAF survey communications. The University of Alabama at Birmingham Institutional Review Board approved this secondary analysis of the PAF survey data.”

3. Thank you for stating the following in the Competing Interests section: "I have read the journal's policy and the authors of this manuscript have the following competing interests: Dr. Rocque is supported by an American Cancer Society Mentored Research Scholar Grant (MRSG-17-051-01-PCSM) and has received research funding from Genentech, Pfizer, and Carevive and consulting fees for Genentech and Pfizer." We note that you received funding from a commercial source: Genentech. Please provide an amended Competing Interests Statement that explicitly states this commercial funder, along with any other relevant declarations relating to employment, consultancy, patents, products in development, marketed products, etc. Within this Competing Interests Statement, please confirm that this does not alter your adherence to all PLOS ONE policies on sharing data and materials by including the following statement: "This does not alter our adherence to PLOS ONE policies on sharing data and materials.” (as detailed online in our guide for authors http://journals.plos.org/plosone/s/competing-interests). If there are restrictions on sharing of data and/or materials, please state these. Please note that we cannot proceed with consideration of your article until this information has been declared. Please include your amended Competing Interests Statement within your cover letter. We will change the online submission form on your behalf.

Response: We have edited our Competing Interests Statement to the following: “I have read the journal's policy and the authors of this manuscript have the following competing interests: Dr. Rocque is supported by an American Cancer Society Mentored Research Scholar Grant (MRSG-17-051-01-PCSM) and has received research funding from Genentech, Pfizer, and Carevive and consulting fees for Genentech and Pfizer for work unrelated to the current study. This does not alter our adherence to PLOS ONE policies on sharing data and materials.” We have also included this in our updated cover letter.

We have edited our data availability statement to the following: “Data that support the findings of this study was collected for organizational and programmatic purposes by Patient Advocate Foundation and may be available upon request. Participants of this study did not agree for their data to be shared publicly.”

Response: We have reviewed our reference list and confirm it is complete and correct.

Response to reviewer comments:

Reviewer 1

This is a well written and important topic on financial burden for cancer patients in the United States during the pandemic. Overall this is a well thought out paper, but I do have a few significant suggestions, and one minor comment.

Significant

1. The authors have made no attempt to identify how the financial impact to patients has changed since the pandemic. I am not suggesting additional analysis, but rather an examination of data pre-pandemic to see if the impact is different (or the same) as it was pre-pandemic in the discussion or limitations. 

Response: We agree this is an important issue to address and have added to our discussion section using data from Han and colleagues (https://doi.org/10.1158/1055-9965.EPI-19-0460). 

“Our results suggest the financial hardship experienced by cancer survivors may have been exacerbated during the COVID-19 pandemic when compared to pre-pandemic estimates, such as those in a recent study by Han and colleagues. Using the 2016 Medical Expenditures Panel Survey data, Han estimated 16% of cancer survivors experienced material financial hardship, which includes reduced spending on non-healthcare needs, and 27% experienced delayed or forgone health care due to cost.” 

2. The authors have not addressed the overall impact on employment during the pandemic. Although some federal funding was provided in many cases this would have been a decrease over their regular income stream. Again I am not suggesting additional analysis but at a minimum this needs to be highlighted in the limitations, or an examination of the literature on this topic should be included in the discussion.

Response: We appreciate this very relevant suggestion. COVID-related employment changes were uncommon in our study, with 2% of respondents self-reporting pandemic-related loss of employment. Notably, 65% of our sample reported being retired or disabled at the time the survey data was collected, which suggests most of our sample would not be affected by pandemic-related employment changes. Furthermore, for minimum-wage earners in our sample, receipt of unemployment benefits may have also resulted in income increases rather than decreases. We have added these details to our limitation section. 

“The results of our study could have been impacted by pandemic-related employment changes. However, this was uncommon in our study with only 2% of respondents reporting employment loss due to the COVID-19 pandemic.“

Minor

3. pg 12 of 21 line 122 "which covers may oral..." should read "which covers many oral..."

Response: Thank you for the attention to detail. We have edited accordingly.

Reviewer 2 

This is a well-written and concise analysis, and it provides a meaningful addition to the growing (and necessary) body of literature on the impact of COVID-19 on cancer patients and survivors. I particularly appreciate your including cost of internet in your survey, as this is, as you note, a potential barrier to telehealth uptake and may limit survivors' ability to engage with other aspects of the healthcare system as well. My comments are minimal but include the following:

1. Please consider omitting your use of "under-resourced" throughout the manuscript, or at least provide a clear definition to understand the research team's characterization of this term.

Response: Thank you for this suggestion. We have deleted the term “under-resourced” from our manuscript. We have also clarified the potential sample bias in our limitations. 

“The survey captures data from cancer survivors who previously received help accessing or paying for care from PAF and may not be representative of the larger cancer patient population.”

2. Did you collect any information on time since treatment or diagnosis, or on the respondents' current healthcare use/needs? If so, please report; if not, please note as a limitation, as these factors, particularly current healthcare needs, may influence one's difficulty paying for healthcare use.

Response: We agree that both time since diagnosis and current healthcare use or needs could influence payment for both healthcare and non-healthcare needs. However, the survey data used for this study was collected by Patient Advocate Foundation for administrative and programmatic purposes. Our analyses were thus limited by the use of secondary data. We have added this to our limitations section.

“Information potentially associated with both health insurance coverage and challenges paying for healthcare and non-healthcare needs, such as time since cancer diagnosis, current health care use or needs, and more detailed demographic information, was limited by use of secondary data.” 

3. Please provide more detail on how your outcomes of interest were measured? Were these yes/no questions, or did you use a scale? If the latter, provide detail on how the scale was dichotomized.

Response: Respondents reported challenges paying for healthcare and non-healthcare needs using the single-item survey question, “Have you had trouble paying for any of the following since the beginning of the COVID-19 pandemic?” Respondents were then asked to select all that applied, which included food, household supplies, housing (rent or mortgage), utilities, phone, internet/data, car/gas/transportation, childcare/eldercare/home health services, and healthcare/medical costs (prescription medications, doctor’s visits, clinical/hospital services, medical supplies, etc.). We have added these details to our methods section. 

4. Was age collected as a continuous variable, or was it collected in the categories reported in Table 1? If the former, consider revising your categorizations more meaningfully to reflect age 65 as the age of Medicare eligibility and 18-39 as the NCI's definition of young adult. Similarly, how was the income threshold of $48K determined?

Response: Age data was captured using the categories reported in Table 1, which again points to limitations presented by use of secondary data. We have added this to our limitations section.

“Information potentially associated with health insurance coverage and challenges paying for healthcare and non-healthcare needs, such as time since cancer diagnosis, current health care use or needs, and more detailed demographic information, was limited by use of secondary data.” 

The income data in our study, which was also collected categorically, naturally fell into quartiles. To better understand potential differences between respondents with higher and lower household incomes, we dichotomized this data into those who had annual household incomes above and below the third quartile of income data. This translated to those an annual household income of < $48,000 (74% of our sample) compared to ≥ $48,000 (25% of our sample). Detailed income data is included in the table below.

Annual household income n %

≤ $23,999 486 33.6

$24,000-$47,999 582 40.2

$48,000-$71,999 223 15.4

$72,000-$95,999 74 5.1

$96,000-$119,999 13 0.9

≥ $120,000 52 3.6

Unknown 17 1.2

Thanks again for your consideration.

Sincerely,

Courtney P. Williams, DrPH 

Postdoctoral Fellow

---

## [Decision Letter · Decision Letter 1]

26 Jul 2022

Health insurance and financial hardship in cancer survivors during the COVID-19 pandemic

PONE-D-21-32975R1

Dear Dr. Williams,

We’re pleased to inform you that your manuscript has been judged scientifically suitable for publication and will be formally accepted for publication once it meets all outstanding technical requirements.

Kind regards,

Raymond Nienchen Kuo, Ph.D

Academic Editor

PLOS ONE

Reviewers' comments:

Reviewer's Responses to Questions

**Comments to the Author**

1. If the authors have adequately addressed your comments raised in a previous round of review and you feel that this manuscript is now acceptable for publication, you may indicate that here to bypass the “Comments to the Author” section, enter your conflict of interest statement in the “Confidential to Editor” section, and submit your "Accept" recommendation.

Reviewer #1: All comments have been addressed

Reviewer #2: All comments have been addressed

2. Is the manuscript technically sound, and do the data support the conclusions?

Reviewer #1: Yes

Reviewer #2: Yes

3. Has the statistical analysis been performed appropriately and rigorously? 

Reviewer #1: Yes

Reviewer #2: Yes

4. Have the authors made all data underlying the findings in their manuscript fully available?

Reviewer #1: No

Reviewer #2: Yes

5. Is the manuscript presented in an intelligible fashion and written in standard English?

Reviewer #1: Yes

Reviewer #2: Yes

6. Review Comments to the Author

Reviewer #1: I am satisfied with responses from the authors. I have no further questions for the authors. Well done.

Reviewer #2: Thank you for your attention to addressing my previous comments and those of the other reviewer. I have no additional comments.

7. PLOS authors have the option to publish the peer review history of their article (what does this mean?). If published, this will include your full peer review and any attached files.

Reviewer #1: **Yes: **Christopher J. Longo

Reviewer #2: No

---

## [Editor Report · Acceptance letter]

28 Jul 2022

PONE-D-21-32975R1 

Health insurance and financial hardship in cancer survivors during the COVID-19 pandemic 

Dear Dr. Williams:

I'm pleased to inform you that your manuscript has been deemed suitable for publication in PLOS ONE. Congratulations! Your manuscript is now with our production department. 

Kind regards, 

on behalf of

Professor Raymond Nienchen Kuo 

Academic Editor

PLOS ONE